# The Longevity of Colonies of Fungus-Growing Termites and the Stability of the Symbiosis

**DOI:** 10.3390/insects11080527

**Published:** 2020-08-13

**Authors:** Margo Wisselink, Duur K. Aanen, Anouk van ’t Padje

**Affiliations:** Laboratory of Genetics, Wageningen University, Droevendaalsesteeg 1, 6708 PB Wageningen, The Netherlands; margo.wisselink@wur.nl (M.W.); anouk.vantpadje@wur.nl (A.v.P.)

**Keywords:** *Termitomyces*, horizontal transmission, mutualism, mutation, cheaters

## Abstract

**Simple Summary:**

Fungus-growing termites cultivate monocultures of a specific fungi (of a genus called *Termitomyces*) for food in their colony, analogously to human farmers growing crops. The termites forage for dead plant material in the environment, bring this into the mound and provide it to the fungus as a growth substrate. After fungal growth, the termites use the mixture of fungus and degraded plant material and also the asexual spores produced by the fungus as food. They also use the spores to inoculate new fungus gardens. The termite-fungus symbiosis is obligatory for both partners, the termites provide growth substrate and a protected growth environment for the fungi in exchange for a nitrogen-rich food source. Termites and fungi have been partners for more than 30 million years and although the symbiosis looks harmonious, the partners have some hidden conflicts about reproduction and resource investment. These conflicts raise questions on how the symbiosis apparently has remained stable for such a long time. We summarize our current understanding on the short-term stability of a single generation of termites, and also the long-term stability of the symbiosis across multiple generations. We identify unsolved questions in the field and suggest possible avenues to address these questions.

**Abstract:**

The agricultural mutualistic symbiosis between macrotermitine termites and *Termitomyces* fungi is obligate for both partners. The termites provide a protective growth environment for the fungus by cultivating it inside their colony and providing it with foraged plant material. The termites use the fungus for plant substrate degradation, and the production of asexual fruiting bodies for nourishment and re-inoculation of the fungus garden. The termite colony can reach an age of up to several decades, during which time it is believed that a single fungal monoculture is asexually propagated by the offspring of a single founding royal pair. The termite-fungus mutualism has a long evolutionary history dating back more than 30 million years. Both on the time-scale of a termite colony lifespan and that of the mutualistic symbiosis, questions arise about stability. We address the physical stability of the mound, the termite colony and the monoculture fungal garden during a colony’s lifetime. On the long-term evolutionary scale, we address the stability of the symbiosis, where horizontal transmission of the symbiotic fungus raises the question of how the mutualistic interaction between host and symbiont persists over generations.

## 1. Stability of Symbioses

Stability is the quality or state of being steady and not likely to change, fail or decay [1]. In ecosystems, stability includes both the ecosystem’s resistance to perturbation, and the speed with which a perturbed ecosystem returns to its original state [1]. This definition can be stretched to be used for symbiotic relationships, when two or more different species associate in a long-term relationship. For example, the adult human microbiome, the microbial community in and around the human body, can sustain major perturbations [2], showing a long-term stable association between the human body and the microbiome. 

Host-microbe symbioses involve two or more differently sized species, the host, defined as the larger species, and the symbiont, the smaller species. The smaller sized symbionts are often found to cooperate in a group with the host [3,4]. Often, the group of symbionts collectively produce goods and/or services for the host, while the host provides the symbionts with a safe environment, enhancing all participants’ inclusive fitness [4]. Generally, the host tends to be favored by an optimization of the collective performance of the symbionts, and this often depends on cooperation among the symbionts to reach maximum yield. The understanding of both the maximization of the collective performance of symbionts by the host, and conflict reduction between both partners seems key for explaining the stability of the symbiosis [5]. 

The stability of several symbiotic relationships has been studied, both in theory [6,7,8] and experimentally [9]. For example, arbuscular mycorrhizal fungi have been stably colonizing plant roots for the last 450 million years [10], the coral *Plexaura kuna* retains the same zooxanthellae algal symbionts for over 10 years, despite elevated water temperatures [9], and certain fungus-growing termites have been in symbiosis with specific fungi for more than 30 million years [11,12]. 

This review focuses on the evolutionary stability of the obligatory mutualistic symbiosis between the fungus-growing termites (subfamily Macrotermitinae) and their fungal symbionts (genus *Termitomyces*), on the longevity and the short-term stability of single colonies of different termite species, and on the long-term stability of the symbiosis. This symbiosis is a good model system to study stability, since it is very old and obligatory mutualistic for both partners [11,13,14,15]. The Macrotermitinae have been divided into approximately 330 species over eleven genera and are only found in the old world tropics (Indomalaya, Arabia and Africa) [16,17]. Approximately 50 *Termitomyces* species have been described [18]. In Africa, termites have major effects on the environment at a landscape level. Termites move soil particles, water and nutrients and have an effect on the chemical and physical properties of the soil. Therefore termites have been called ecosystems engineers [19]. Termites cultivate the fungus on a fungus comb built from foraged plant material [20], while showing a complex age polytheism in this process [21,22,23]. The older termites undertake foraging activities and transport the plant material to the nest. Younger termites mix this plant substrate with *Termitomyces* asexual spores in their guts, this mixture is deposited as the fungal comb. The fungus decomposes the plant-substrate and forms asexual fruiting bodies, which are eaten by the younger termites for nutrition and re-inoculation of the fungus garden. When the comb is fully decomposed, the old comb is consumed by the older termites for nourishment [20,24]. The termite colony can reach an age of up to several decades [16], although longevity has not been studied extensively. 

The fungal symbiont is acquired from the environment via sexual spores (Figure 1). When the colony is established, it is likely that a mixed culture of multiple different heterokaryotic fungal colonies is formed. However, mature colonies contain a single heterokaryon clone of *Termitomyces*, indicating that termites cultivate their fungi as monocultures [25,26]. Monocultures are most likely established and maintained through positive frequency-dependent selection, a process through which the most abundant fungal genotype is propagated by the termites. Additionally, during inoculation, the fungus will repeatedly face bottlenecks, which contributes to the reduction of the number of clones [26,27]. 

The last years have seen new publications on evolutionary and ecological patterns in the symbiosis between termites and *Termitomyces.* These findings raise many new questions on stability, both on the short-term, the lifetime of the termite colony and the long-term, the evolutionary history of the mutualistic symbiosis. The recent development of establishing lab colonies for experimental work, opens up the possibility for controlled lab experiments to address some of those questions [24]. In this review, we explore the physical stability of the mound, the longevity of the termite colony and the stability of the monoculture fungal garden. We address the current knowledge on these topics and to compile a list of open questions in the field. Furthermore, we discuss further research possibilities in the field. For example, experimental evolution and mutation-rate estimation in other fungi show variable mutation rates per generation [28,29,30], which can cause the rapid appearance of mutants when sequentially transferred [31], raising the question how *Termitomyces* keeps mutations in check during long-term clonal monoculture cultivation. On an evolutionary scale, we address the stability of the mutualistic aspect of the symbiosis, where the mutualistic interaction between host and symbiont has to be persistently maintained over the generations, even though horizontally transmitted.

## 2. The Stability of the Termite Mound

### 2.1. The Physical Structure of the Termite Mound

A colony of termite species of the genus *Macrotermes* occupies of a large above-ground earth mound centered on a below-ground gallery system, connected with tunnels and air ducts [19]. The earth mounds can superficially be seen when the colony is already well-established and mature. When the colony gets older, the mound gets bigger and can reach a height of up to 10 m with diameters of over 15 m [19,32]. 

#### 2.1.1. Erosion and Persistence

Many young and old termite colonies are destroyed through predation, disturbance, competition, and possibly disease [33]. The termite mound protects the colony from these factors by providing protection against predators and hostile environments [34]. The termites mix soil particles of specific sizes with clay and saliva to create mounds with a hard exterior resistant to predators and desiccation [35,36]. The mound is constructed with tunnels and air ducts to regulate temperature and moisture levels [14]. Torrential rain and other factors erode the termite mound continuously; however, termite workers continuously work to repair damages and breaches keeping the mound stable [37]. When no longer occupied by termites the mound is susceptible to erosion. The erosion of mounds is complex and influenced by many factors [32,33]. The mounds are susceptible to destruction by animals and humans, the incidence of fire, floods and weather influences. However, the mounds colonies can persist when the termite colony is no longer active, that is to say, longer than the lifespan of a single termite queen. When termite colonies decease, the earth mounds can stay stable and persist as landscape features because of protection by the vegetation cover that stabilizes and shelters the mound or constant recolonization by successive colonies [32,33]. 

#### 2.1.2. Reuse by New Termite Colonies

It has previously been suggested that large earth mounds examined can be the result of one termite colony building, but it is more likely they are the result of a long duration of occupation of several termite colonies and even different termite species [32,33]. This is supported by previous observations, where large earth mounds were found with the active termite colony and the active brood chamber located well above ground level, whereas abandoned, non-active, gallery systems were located below-ground [32,33]. One explanation would be that when a new termite colony is established, the abandoned eroding mound is perceived as a good site to colonize. Alternatively, colonization could be random, but there is a higher survival of new colonies in existing mounds. It has been suggested that the large size of the earth mounds can be the result of successive building and erosion occurring because of abandonment and recolonization [32]. The size of the earth mounds would not be achieved during the 20 years lifespan of one termite queen and king [32,38,39]. Repopulation of abandoned mounds is also found in the earth mounds of the non-fungus growing termite species *Microhodotermes viator*, locally called *Heuweltjies*. Research reveals evidence of earlier inhabitation of the lower portions of the mounds by the same species [40].

#### 2.1.3. Ages of Termite Mounds

Estimated ages of termite mounds vary between species. Using skeletal material from an ancient burial ground, preserved inside the alkaline soil of the termite mound, the age of a *Macrotermes falciger* mound was estimated at about 700 years [41]. Dating of the basal part of the *Heuweltjies*, earth mounds of *M. viator*, indicates an existence for at least 4000 years [40]. *M. viator* has occupied the mounds throughout this period, demonstrated by trace-fossil evidence. In theory, in lower termites as *M. viator,* all castes can reproduce, meaning that the termite colony has the potential for indefinite lifetime [33,40]. Since the earth mound has survived this many years, it is probably not left unoccupied for a long time, if at all, since erosion and other forces did not destroy it in this extended period [40]. With this study, it was not possible to determine whether the termite colony changed over time, or that the mound was inhabited by one single termite colony continuously. The termite origin of these *Heuweltjies* has been contested in more recent studies [32]. Dating along the central vertical axis of a largely abandoned earth mound of *M. falciger*, unquestionably termite-built, indicates an existence for at least 2200 years [32].

### 2.2. The Longevity of a Termite Colony

Recently, it was discovered that in some termite species, secondary queens are produced by parthenogenesis of the primary queen, so-called asexual queen succession (AQS) [42]. By cloning herself, the queen can extend her (genetic) lifespan. An additional consequence is that the queen can increase her genetic contribution to offspring beyond the 50% that results from normal reproduction. The reason is that the primary king cannot produce secondary kings by parthenogenetic reproduction, but only by mating with the queen. Consequently, the secondary kings are 50% (or more, if there have been multiple rounds of replacement) related to the queen, and hence, the offspring is 75% related to the queen.

It had been known for a while that in some termite groups, the primary reproductives can be replaced by secondary reproductives, thus prolonging a colony’s lifespan [43]. Stephen Bartz [44] hypothesized that reproduction by secondary queens and kings was important for the evolution of eusociality in termites. Replacement reproduction would lead to inbreeding, but occasional outbreeding between dispersing alates would then generate high relatedness between offspring favoring them to help the queen and king produce more siblings rather than producing offspring themselves. Close analysis of Bartz’ theory has demonstrated that the assumptions and predictions are not fulfilled in most termite groups [43]. Most importantly, replacement reproduction is not universal across termites. None of the Macrotermitinae have been reported to have replacement reproduction, neither by inbreeding between secondary reproductives, nor by asexual replacement. So the lifetime of a colony of fungus-growing termites is limited by the maximum life span of the primary reproductives, which has been estimated at a maximum of 20 years [38,39]. This is different from the lower termites, where all castes can reproduce, meaning that, termite colony (or colonies) can extend their lifespan, although this can have external limitations, in particular in species that live inside their own food source, such as many wood-feeding termites.

## 3. How Short-Term Stability of the Symbiont May Affect Long-Term Stability of the Symbiosis

The management of the fungus garden affects both the short-term and the long-term stability of the mutualism, as we will discuss in this section. There is a conflict of interest between the two partners, but also between the symbiont if multiple symbionts are housed together. By keeping and maintaining a monoculture of symbionts, the conflict between fungal symbionts is reduced. Keeping a monoculture for multiple decades may cause problems with the genetic stability of the monoculture when mutations occur. The solution to one problem, of conflict between different fungal symbionts by growing a monoculture, may thus create another problem: lower-level selection within a monoculture. 

We first discuss how the termite and the fungal symbiont disperse and how the fungal symbionts are transmitted to the next generation of termites. Horizontal transmission of symbionts has pros and cons, and in the second part of this section, we discuss how the “right” partners find each other. We then discuss how horizontal transmission leads to direct and indirect conflicts of interest in resource investment between termite host and fungal symbiont, and how this conflict is reduced. Lastly, we discuss how fungal monocultures are established and maintained by the termites to increase the spore yield by reducing conflict between fungal symbionts.

### 3.1. Symbiont Transmission and Dispersal

The symbiosis between termites of the Macrotermitinae and fungal species of the genus *Termitomyces* dates back more than 30 million years [11,12]. The symbiosis is obligatory for both partners [16,45], and reversals to a non-symbiotic state are not known, neither for the termites nor for the domesticated fungi [16]. 

In most genera of Macrotermitinae, the fungal symbiont is horizontally transmitted to the next generation of termites, meaning, that the winged alates do not carry the fungal symbionts with them when dispersing during the rainy season (Figure 1). After swarming, the winged alates form a pair, build a copularium and produce new worker termites which form a primordial comb. This primordial comb needs to be inoculated with the fungal symbionts for the termite colony to thrive. After the establishment of a new colony, the termites thus have to acquire a new fungal symbiont from the environment [13,16,45]. On one of their first foraging trips, the new termite workers bring sexual spores of *Termitomyces* into the primordial comb [13].

Fungal spores can be found in the environment because many species of the horizontally transmitted *Termitomyces* produce above-ground fruiting bodies every year. These fruiting bodies release large numbers of sexual wind-dispersed spores into the environment [46] (Figure 1). The production of fruiting bodies on the termite mounds is a seasonal event and correlates with rainfall, the nuptial flights of the alates and the appearance of new termite workers [47,48]. The new workers appear when the *Termitomyces* of related, mature, termite colonies are fruiting. Depending on the *Termitomyces* species, this occurs one-and-a-half to three months after the swarming event [47]. The exact mechanisms behind this correlation remain to be discovered, but might be induced by behavioral changes of the termites after the swarming event or the sudden drop in food consumption in the termite mound when the alates have left the colony [27,47,49].

The synchronized timing of fungal fruiting and the appearance of termite workers from newly established colonies helps those to acquire fresh fungal spores from the environment to inoculate their fungus garden. It also helps the fungus to disperse into new termite colonies. Co-evolution between termites and fungi might have favored the timing of fruiting and alate dispersal.

Horizontal transmission has its pros and cons. The independent reproduction of termites and fungi [16], and the horizontal transmission mode of symbionts gives the termite the flexibility to acquire a different fungal symbiont at each generation [50]. This is a form of partner selection, and gives termites a time window to select the best performing fungal symbiont. However, how do partners find each other? The horizontal transmission also allows the symbiont to disperse away from close relatives and avoid competition [3]. However, the horizontal transmission also creates a conflict between the two partners in the symbiosis.

### 3.2. Selection of the “Right” Partner

There is some interaction specificity between fungus-growing termites and their fungi, but this varies between species [51,52]. When comparing host and symbiont phylogenies, it has been shown that the interaction specificity was much higher at the genus level than at the species level [52], indicating that certain groups of termites have coevolved with certain groups of symbionts. It is still an open question of how this specificity arises and how specific host-symbiont combinations arise [52].

The observed host-symbiont combinations might be correlated with environmental factors and might be the effect of spatial autocorrelation [52]. Recent data shows that the structure of the termite mound, and hence, the temperature, pH and vegetation type has some correlation with specific *Termitomyces* species, rather than with the termite species living in these mounds [51,53]. It is known that some environmental factors, for example, the carbon source in the substrate, can affect the growth of *Termitomyces* [54]. Carbon source could, thus, theoretically affect the competitive strength when the symbiont has to compete with other *Termitomyces* species. While it is unknown how many *Termitomyces* species are brought into the primordial comb by the first termite workers, environmental factors could, theoretically, explain why only one fungal species remains. The *Termitomyces* species that grows best and produces most asexual spores in that particular environment will outcompete other *Termitomyces* species. Disturbances and changes in the substrate might have a smaller impact on *Termitomyces* species which are metabolically flexible and are able to digest a wider range of carbon sources. Such metabolic flexibility would, under changing conditions, outcompete other *Termitomyces* which do not have this flexibility. However, under stable conditions, metabolic flexibility might be costly or less efficient than degrading, only one type of carbon. The degree of metabolic flexibility in *Termitomyces* and how much the growing conditions can change during the lifespan of a termite mound remain unknown. To test these ideas, research should investigate whether these environmental factors vary over time between the termite colonies of the same species and whether this can affect the establishment of a particular fungal symbiont. It is also unknown to what extent the termite or fungal species can determine these conditions (Appendix A). 

It also remains an open question if the primordial comb ever contains more than one fungal species, or whether the selection of the “right” fungal symbiont happens before the spores are brought into the mound, either actively or passively. Since the timing of fruiting of *Termitomyces* and the appearance of the first termite workers are correlated, one specific *Termitomyces* species might be disproportionally abundant in the environment. Without any active selection, the termite workers might pick up that particular *Termitomyces* species which grows well in that particular environment [47]. It is also possible, however, that termites more actively select a particular *Termitomyces* species. Recent research has shown that termites are able to distinguish the pathogen *Pseudoxylaria* from their mutualist *Termitomyces* by smell, through emitted volatiles [55]. Behavioral studies on termites should determine if the termites actively select “their” fungal symbiont—the symbiont they have co-evolved with—over other *Termitomyces* fungi. It has been shown that different *Termitomyces* species have qualitative differences in their volatile profiles [55]. Termites might react differently to co-evolved and not co-evolved *Termitomyces* species, favoring one species over another. It remains an open question if the termites actively select one fungal species over the other, and if so, what the mechanisms behind this selection would be.

### 3.3. Conflict Reduction between Termite Host and Fungal Symbiont

Obviously, sexual reproduction and horizontal dispersal of the fungus and production of alates by the termite colony are crucial for the persistence of the symbiosis. However, there is a direct and an indirect conflict of interest between partners about resources invested in dispersal (Table 1). In this section, we describe this fundamental conflict between the termite and the fungus over the resources allocated to dispersal and how this can be minimalized. 

The reproduction of termite hosts and fungal symbionts is decoupled: The dispersal of the fungal symbiont is not dependent on the dispersal of the termites and vice versa, which can lead to a conflict of interest over the resources allocated to reproduction. The termite does not have a direct interest in the sexual reproduction of its fungal symbiont, and could therefore, be selected to prevent the fungus from investing resources into fruiting bodies. The termites, thus, might have evolved mechanisms to suppress the production of fungal fruiting bodies, for example by managing the fungal growth, because selection would favor termites which can repress fruiting bodies and have more fungal resources for food. The fungus, on the other hand, does not have a direct interest in the production of winged alates and the dispersal of the termites, because when alates are formed less energy is invested in foraging for plant materials or building more fungal combs, inevitably decreasing the growth potential of the fungus. The fungus would instead rather force the termites to invest energy in building more fungal combs [14,45]. It is known that some other fungi are able to affect the behavior of the host to increase their own reproduction; for example, basidiomycete fungi of the genus *Fibularhizoctonia* mimic the eggs of the termite *Reticulitermes speratus* to be quickly transported to new competitor-free areas within the termite nest [56]. However, whether *Termitomyces* has evolved mechanisms to affect termite behavior is not known (Appendix A).

Theory shows that symbiotic partners also have an indirect conflict of interest about dispersal. Symbionts are favored to disperse away from their close relatives to avoid competition with those [57]. However, horizontal transmission of symbionts can lead to symbiont mixing in the host. Symbiont mixing can lead to greater within-host competition for resources and increased virulence against the host, decreasing the host’s fitness. When genetically different symbionts are present, the success of a symbiont genotype depends relatively more on its competitive success within that group than on the overall success of the group [3]. Symbionts with increased antagonistic properties, have increased competitive success at the expense of reproduction and the success of the group [26,27]. Genetic variation within a group of symbionts could, therefore, lead to the selection of symbionts with higher competitive strength, but lower benefits to the host [58]. To reduce this effect, it is in the host’s interest to avoid symbiont mixing, and this is even more so if there is a direct negative effect of symbiont mixing [3]. Since the symbiosis between fungus-growing termites and *Termitomyces* is characterized by horizontal transmission symbiont mixing could occur and fungus-growing termites would, thus, benefit from restricting the number of fungal symbionts, to avoid selection for more virulent characteristics.

### 3.4. Conflict between Symbionts: Establishment and Maintenance of Fungal Monocultures

The sexual spores of *Termitomyces* brought into the primordial comb are homokaryotic, and when germinated, will form homokaryotic mycelium (Figure 1). When two homokaryotic mycelia meet, they can fuse and form a heterokaryotic mycelium, if they are sexually compatible (i.e., if they have different mating types and belong to the same biological species) [26,59,60]. Even though horizontal transmission of the fungal symbionts would allow for the establishment of multiple genetically different heterokaryons within one termite mound, all screened colonies did not contain genetic variation, and thus, only a single heterokaryon from a single *Termitomyces* species was associated with a single colony [16,25,26,51,59,61]. Other research on *Termitomyces* has shown that the number of nuclei per cell in the heterokaryon is variable, but that it are always only two different nuclei, meaning that only two homokaryons pair to form one heterokaryon, never more [59]. 

Growing monocultures means that there is no direct competition between the fungi, due to symbiont mixing, and monocultures are ecologically stable and maintained by positive frequency-dependent selection (see below). Genetic uniformity among the fungal symbiont is critical because when multiple symbionts are present in a termite mound, the symbionts spend resources in competing with each other instead of growth, with as a result lower asexual spore yields. It is, thus, beneficial for the termites to establish and strictly maintain a monoculture because it is the key to the highest yield [26]. However, farming a monoculture also may make a culture susceptible to parasites and diseases [62], decreasing the yield [63] and might, in severe cases, even cause a colony to collapse. In particular, species of the ascomycete genus *Pseudoxylaria* are known to compete with the *Termitomyces* cultivar [55,62,64]. Therefore, termites have developed ways to recognize pathogens and to effectively remove these from the fungal comb [55,65,66]. The process of removal of diseases and parasites is, thus, essential for the maintenance of healthy fungal monocultures and the stability of the symbiosis. While the mechanisms behind the establishment and the enforcement of a monoculture are still not completely understood [53], recent work has tried to unravel the underlying forces. The monocultures are likely established and maintained by positive frequency-dependent selection, possibly in combination with recurrent bottlenecks.

#### 3.4.1. Positive Frequency-Dependent Selection and Bottlenecks

Once growing on the fungal comb, *Termitomyces* produces asexual spores which are used by the termites as food and to maintain their fungal garden (Figure 1) [23,27]. The asexual spores pass the guts of the termites, and only a small fraction of asexual spores will be propagated to further inoculate new fungal combs in their mound.

It has been demonstrated that the monoculture can be established and maintained by positive frequency-dependent selection (Figure 1). The frequency of a dominant fungal strain will increase, and over time the rarer fungal strains will be lost from the colony altogether [26]. This positive frequency-dependent selection presumably occurs by fusion among clonally related fungal strains: Fusion leads to the formation of larger networks and higher production of asexual spores [67]. Additionally, since asexual spores are used for intranest transmission, continuous bottlenecks occur, further reducing genetic variation [27]. 

While likely both positive frequency-dependent selection and bottlenecks are important mechanisms in the maintenance of a fungal monoculture, many questions remain unanswered [53]. Future research should focus on these exact mechanisms (Appendix A). 

#### 3.4.2. Symbiont Turn-Over

While never more than one *Termitomyces* strain has been observed within one termite mound, it remains an open question if there ever occurs fungal turnover within the fungal mound. It has been hypothesized that such fungal turnovers never occur, since it will be hard for any new strain to invade an established monoculture, since a new strain will per definition start at low frequency. It is this lifetime commitment between the termite and the fungus which has been hypothesized to lead to the evolutionary stability of this symbiosis. The two partners commit to an extreme form of partner fidelity, which immediately removes the incentive for cheating and will favor increased symbiont performance [26]. Restricted effective immigration of new symbionts also decreases the competition between symbionts, and increases the mean fitness of the host population, increasing the selection on termites which strictly maintain a monoculture [3]. So theory predicts that once the termites have acquired a monoculture, they will maintain this culture and would not switch to another fungal symbiont [26]. Termites likely have evolved mechanisms to enforce monocultures, which has been crucial for the observed evolutionary stability of the symbiosis. However, future long-term studies should test this theoretical prediction that the fungi and termites indeed engage in an extreme form of partner fidelity and also look into the stability of the fungal symbiont over time.

#### 3.4.3. Mutations and Cheaters 

Fungal colonies consist of interconnected, cooperating hyphae of totipotent cells which all have the potential to reproduce [45]. However, only a small proportion of the cells will eventually reproduce. At each cell division, mutations can occur, which can give rise to cheating mutants which have the benefits of the cooperation, but not the cost. Theoretically, cheating mutants will outcompete the cooperating cells, and then cooperation would disappear [68]. Theory predicts that high relatedness will facilitate kin selection and prevent the spread of cheaters, because cheaters need to be mixed with cooperators to outcompete those, but groups consisting of too many cheaters will have lower reproductive success than groups of cooperators. Genetic relatedness between cooperating cells, thus, is a fundamental factor to maintain cooperation [58,69]. Experimental evolution has supported this theory and shown that the occurrence of cheating mutants, which do not contribute to the maintenance of a cooperating colony, but only to the reproduction, arise and spread through the colonies when relatedness between the colonies is low, but cannot spread when relatedness is high [58]. 

#### 3.4.4. Termite Adaptations against Fungal Genetic Variation 

Maintaining high genetic relatedness between the fungal symbionts in the termite mound is, thus, essential to prevent competition among the fungal strains, or antagonistic and cheating traits to spread through the fungal population. By keeping a monoculture in the termite mound, the termites minimize fungal genetic variation and increase the yield by immediately reducing competition and on the long-term reducing selection for cheating fungal strains. However, once established a monoculture, genetic variation can arise, due to mutation at each cell division, which can in the absence of selection at the level of the fungal mycelium, lead to the loss of yield. We hypothesise that termites might have evolved additional mechanisms to minimize the accumulation of mutations in their fungal symbiont.

The termites reduce genetic variation within the fungal combs by propagating only a fraction of spores, and thus, inducing bottlenecks. Any variation will be filtered out by the bottleneck: Mutations are either immediately lost in the fungal population or can go to fixation very quickly. How does the termite prevent cheating nuclei from spreading? Propagation of asexual spores is a way to reduce the selection on the most competitive traits. Because *Termitomyces* mycelia have septate hyphae, so cell compartments, and both types of nuclei of the heterokaryon are always present within one cell, and within the asexual spore, one nucleus can never be favored over the other nucleus. This limits selection at the nuclear level, keeping within-mycelium competition low [59].

Mechanisms to prevent the spread of cheating fungal strains, also known as discriminating mechanisms, are known from other symbioses [70]. To prevent cheating symbionts, hosts can sanction cheaters by for example creating a less beneficial environment for cheaters [71], selectively aborting cheating symbionts [72], or preferentially allocating more resources to non-cheaters [73,74]. However, whether termites are able to discriminate between cooperators and cheaters and if any of these mechanisms are employed by termites is still unknown. Such policing mechanisms would require that termites are able to distinguish cheaters and cooperative genotypes, which requires sufficient sequestration between the different types, and functional testing of them. Potential mechanisms employed by the termite might be: (1) To inoculate the fungal comb with low density, this way the termites would have the opportunity to evaluate the growth of each fungal spore individually and evaluate the fitness of each fungal spore before fusion; or (2) via a preferential selection of certain nodules. If termites would prefer larger nodules, and there is a trade-off between nodule size (yield) and growth rate, termites would select cooperating, slowly growing symbionts with larger nodules, reducing the fitness of fast-growing cheaters. Future research should look first, into the scope for cheating in *Termitomyces* mycelia, and second, into the mechanisms employed by the termites to reduce the spread of cheating symbionts.

#### 3.4.5. *Termitomyces* Adaptations against Fungal Genetic Variation 

A low mutation rate may be another mechanism to reduce the likelihood that cheaters arise and selection might, thus, have favored *Termitomyces* with low mutation rates. Mutation rates are known to largely differ between different organisms, mainly depending on the body size, and longevity of the organism [75]. Taller, longer-lived organisms generally have lower somatic mutation rates, slowing down the accumulation of mutations during growth and aging [76,77,78]. 

Some fungi are known to be very long-lived, and to reach ages of thousands of years, living much longer than *Termitomyces* fungi which are farmed for some decades ‘only’ [79]. These very long-lived fungi have surprisingly low mutation rates [29], and likely have mechanisms to reduce the mutation rates to reduce the occurrence and accumulation of deleterious mutations. One hypothesis is that these long-lived fungi reduce the accumulation of mutations by asymmetrical cell divisions: The fungi are growing at the tip, and by some unknown mechanism, leave the copied DNA strand, with potential mutations behind in the sister cell, while the dividing tip cell keeps the original, error-free, DNA template [80,81]. Other mechanisms would be increased accuracy of the DNA repair and replication machinery to reduce the mutation accumulation [29,75,82].

While the above mechanisms prevent the occurrence of mutations, mutations will eventually occur, with as a potential consequence, that cheaters can arise [58,69]. To prevent the spread of cheaters, some microorganisms have tightly coupled toxin-immunity systems, in which the toxins act against non-kin without the immunity to this toxin [83]. Experimental evolution has shown that cooperating cells can quickly evolve resistance against cheaters [84,85]. If and how *Termitomyces* is adapted or might adapt against cheating fungal strains remains unknown.

The exact role of the termite and the fungus in reducing deleterious mutations and mutations with a competitive benefit within the fungal colony, remains unanswered (Appendix A). A long-term evolution experiment, propagating *Termitomyces* asexual spores mimicking the termites’ behavior might give insight it the genetic stability of the fungus. Studying the mutations accumulated over the generations would give insight into the mutation rate per cell division. We hypothesise that *Termitomyces* has mechanisms to reduce the mutation rate to be able to be successfully cultured by termites for extensive periods. Studying the competitive strength of the evolved and ancestral line, looking at the distribution of spores after a round of competition, would give insight in the occurrence of cheating morphotypes, and whether this is likely to play a role in termite mounds.

## 4. Conclusions

This review focused on the evolutionary stability of the mutualistic symbiosis between the fungus-growing termites and the fungus *Termitomyces*. We addressed the longevity and the short-term stability of single colonies and discuss how this mutualistic symbiosis based on horizontal transmission can have persisted for so long, and why natural selection has not favored exploitative characteristics and reversals to a non-symbiotic state. Many questions remain to be answered (Appendix A).

While many stable mutualisms are characterized by vertical symbiont transmission, *Termitomyces* is transmitted horizontally [16]. The association between termites and their fungus is not strictly species-specific [52] and horizontal transmission gives both partners of the symbiosis the opportunity to select the best performing partner [50]. Termites might actively select their fungal symbiont, or the selection might happen passively, because of environmental factors which might theoretically affect the competitive strength of the fungus, and thus, its selection. 

Since reproduction of the symbiotic partners occurs independently, termites and fungi have a conflict of interest over reproduction and dispersal. Termites do not benefit from fungi allocating resources into dispersal and vice versa, and fungi do not have an interest in their host allocating resources in alates. However, symbiont dispersal gives the symbionts opportunity to reduce competition with relatives [3], benefiting the fungal symbiont, but costly for the termite because it might lead to symbiont mixing. If and how the symbiotic partners affect each other’s reproduction remains unanswered (Appendix A).

Although horizontal transmission indeed gives ample opportunity for the establishment of a mixed fungal culture, all evidence suggests that a termite colony is associated with a single fungal heterokaryon [16,25,26,51,59,61]. This monoculture fungus garden is maintained by positive frequency-dependent selection [26], possibly in combination with bottlenecks [26]. Growing the fungus as a monoculture provides a direct benefit to the termites, since mixed cultures have a reduced yield [26,67]. However, monoculture also imposes an increased risk of diseases and parasitism. Termites and fungi must have evolved mechanisms to decrease this risk to maintain a stable symbiosis. Theory suggests that symbiont turnover in a termite mound does not occur, but this prediction should be tested empirically (Appendix A). Within a monoculture, mutations can be a threat to the stability of fungal productivity. During each cell division mutations can occur, and some of those may have a short-term benefit within the fungal colony because they benefit from the cooperation, but do not equally share the costs. However, various mechanisms will reduce the selective scope for cheating. For example, nuclei are limited in their spread through the fungal colony by septa between the cells. Furthermore, asexual dikaryotic spores are used to re-inoculate the fungal comb, which means both nuclei are inoculated, so a cheating nucleus cannot outcompete the other nucleus [59]. In addition, termites might have the ability to discriminate between cheating and non-cheating fungi and thereby limit the spread of cheaters, again the mechanisms termites employ are yet unknown (Appendix A). Next to a limited spread of cheaters, their occurrence may be limited, due to a reduced fungal mutation rate. Many questions about the stability of the mutualism between termites and *Termitomyces* remain to be answered (Appendix A). Long-term field studies and experimental studies in the lab could help to answer those questions.

## Figures and Tables

**Figure 1 insects-11-00527-f001:**
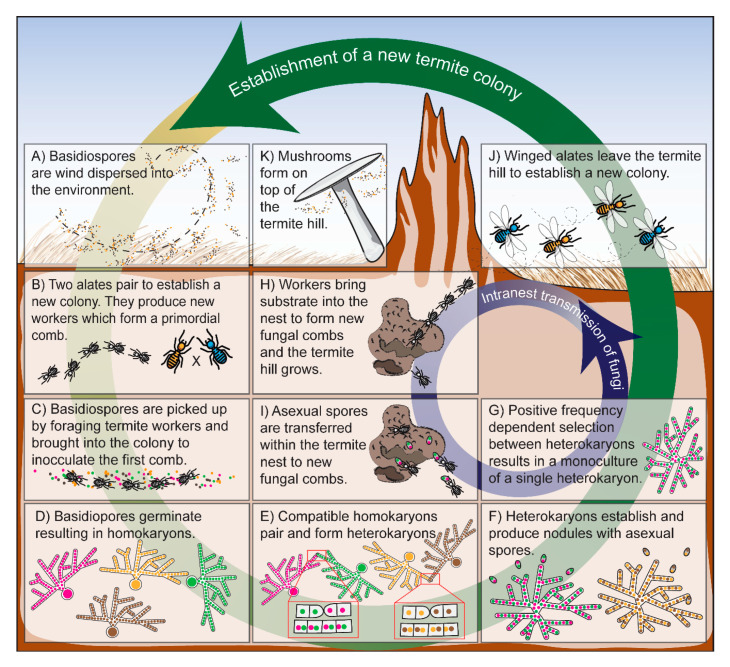
Schematic overview of the establishment of a new termite colony in species with horizontal transmission via sexual spores and the intra-nest transmission of *Termitomyces* via asexual spores. Most species of macrotermitine termites and their fungal symbionts have horizontal symbiont transmission and reproduce and disperse separately. (**A**) Basidiospores originating from *Termitomyces* mushrooms are available in the environment after they have been wind-dispersed. (**B**) Two winged alates become the founders of a new termite colony when they pair and become the king and the queen. The king and the queen produce new termite workers, which form a primordial comb. This comb needs to be inoculated with a *Termitomyces* fungus for the termite colony to survive. (**C**) On one of their first foraging trips, termite workers bring basidiospores of *Termitomyces* into their nest to inoculate the fungal comb. (**D**) The basidiospores germinate and form homokaryotic mycelia. (**E**) When two compatible homokaryons of the same species meet, they can fuse to form heterokaryotic mycelium in the contact zone, with two nuclei in each cell. (**F**) The heterokaryons grow and produce nodules consisting of asexual spores. During the establishment of the colony, it is possible and likely that more than one heterokaryon is present within the termite nest. (**G**) Positive frequency-dependent selection on the different heterokaryons results in the elimination of fungal genetic variation leading to a fungal monoculture. (**H**) Meanwhile, the termites bring new substrate into the nest and build new fungal combs. While the termite colony grows, the termites (**I**) propagate small amounts of asexual spores which are transferred within the nest to new fungal combs. This intranest transmission is a repeating event which continues as long as the termite colony is active. With bottlenecks imposed by the propagation of asexual spores and positive frequency-dependent selection on the most dominant fungal genotype, the fungal mycelium is kept a monoculture, with no fungal genetic variation in the termite hill. (**J**) Seasonally, the termites form winged alates that disperse to establish new colonies. (**K**) Some time after the winged alates have left the colony *Termitomyces* starts forming mushrooms. The fruiting bodies of *Termitomyces* appear on top of the termite hill and disperse large amounts of sexual basidiospores into the environment.

**Table 1 insects-11-00527-t001:** Host- symbiont conflicts in the termite-fungus symbiosis due to horizontal transmission.

Direct	Indirect	
Resources Allocated to Dispersal	Degree of Symbiont Mixing	
Termite and fungus each favor maximum amount of resources for their own dispersal	Symbiont dispersal can lead to symbiont mixing	
	Short-term disadvantage for the termites	Long-term (evolutionary) disadvantage for the termites
	Mixed culture reduces fungal productivity	Competition between unrelated strains selects for competitive traits with virulent side-effects

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
