# Peer review of "The Longevity of Colonies of Fungus-Growing Termites and the Stability of the Symbiosis"

_insects, 2020, doi:10.3390/insects11080527_

Round 1

Reviewer 1 Report

Summary and overview

This manuscript seeks to explore how the mutualistic association between fungus-growing termites and fungal crops persists over the evolutionary time and short-term ecological time. Overall, the review systematizes current knowledge on conflicts between symbiotic partners that underlines the evolutionary stability of this mutualism and also brings pieces of information regarding fungus-growing termite colony longevity. It is a relevant contribution to the field of fungus-growing termite symbiosis, because it teases apart two aspects of the mutualism that ecologists and biologists wonder about. When dissecting these two aspects, the manuscript brings up open questions scattered in the literature and summarizes them in one specific manuscript. The references presented in this study are comprehensive and grasp the major aspects of the theme. The text needs some structural changes to improve its readability (see points #3 and #4). The comments below are mostly centered in the presentation of the text but some are also related to specific scientific issues that are not clear. Hope the suggestions help to improve the manuscript.

Major comments

1. Evolutionary stability of mutualisms is a comprehensive area of research. While is understandable that this manuscript tries to limit the discussion on the target symbiosis (i.e. fungus-growing termites), however, when exploring how mutualistic associations can be stable, there are far more examples (i.e., other fungus-growing insects) available that could be explored in L. 43-56.

2. Although the manuscript brings relevant contribution to the field (see above), this is not clearly stated in the introductory paragraphs. Questions like (i) why this review is timely? and (ii) why is it so important to systematize the literature? are not obvious when reading the introduction section of the manuscript. This review would benefit if the rationale is clearly stated.

3. While it is a nice idea organizing the open research questions in one paper, the manuscript would also benefit if it makes one step forward. Instead of only listing the open questions in Table 2, an extra column should be added in this table to include potential approaches on how to address these questions. There is no need to fill every gap with proposed approaches, but there are approaches that may be available to pursue some of the posed questions.

4. Consider to compile all questions in table 2 and remove them from the main text (kept only when essential). It would also be interesting to clarify which aspects of the termites’ symbiosis suggest these questions as relevant, i.e., considering the ecology of fungus-growing termites, why one should expect such phenomenon to happen? Avoiding the overlapping between the content in table 2 and the main text would render a more straightforward text.

5. By the examples given at the intro, it sounds like the manuscript would also address the resilience of the colonies as an ecological entity, i.e., how termites’ colonies defend themselves from pathogens and predators, survive to environmental variables (as variations in food availability, temperature, humidity) during a colony or population life-span? Thus, it would be interesting to better delimitate the concepts of “short-term” resilience and “long-term” stability, and also explain how these concepts are applied to termites’ symbiosis. Differentiating: i) short-term resilience as adaptations of the colony to recover from perturbations as antagonists, competitors, and environmental variations, and ii) long-term stability as the stability of the mutualism on an evolutionary scale, exemplifying these definitions properly, would probably improve the text clarity.

6. It would be interesting to link the resilience of the mound physical structure to defensive advantages to the colony life-span (short-term resilience), as well as which features of the mound could confer such advantages.

7. The conclusion section should close all gaps highlighted throughout the text, instead of only summarize the information already provided. As it is, the conclusion has several overlapping information with previous sections, but does not seem to recapitulate important aspects (as contextualizing resilience and stability) and to close the thought built during the review.

8. It would be insightful to speculate how Termitomyces performance on different substrates could be correlated with short term resilience, as well as to link the fungal metabolic capacity to the conidia selection by workers. As such, it would be interesting to discuss whether strains that show a metabolic flexibility (i.e., are able to degrade multiple substrates and to survive by metabolizing several carbon sources) would/would not render a more stable association in short and long term. Also, it should be considered to conceive the possible influence of the microbiota on colonies metabolism and nutrition, and how it could affect the stability of the symbiosis.

9. Also very interesting would be to discuss the importance of resilience and stability to fungus-growing termites’ ecology by comparing them to other termites (non-fungus growing) and/or other fungus-growing insects. This would allow the authors to theorize possible impacts of the symbiosis on macrotermitine stability and resilience, and to relate this to the termites life-style and environment.

Specific comments

The title seems not to cover the two aspects discussed in depth in the manuscript. Perhaps, something like “Stability of the mutualism and longevity of colonies of fungus-growing termites”.

Consider use the term "conidia" for asexual spores, throughout the text.

L. 21-22: Consider removing “fungus-growing termites” and “stability” because these keywords also appear in the title.

L. 24: remove “Introduction” and keep only one section heading.

L. 30: replace “inertia and resilience” to “these definition” to avoid repetition.

L. 35-38: The explanation on host-symbiont association is not very clear.

L. 43-56: correct for “P. kuna”.

L. 59: How horizontal transmission occurs in the macrotermitine symbiosis is not clear at this point. It is indeed detailed below, but could cause confusion whether is not clearly stated at the introductory section.

L. 75: It would be helpful to clarify what is “positive-frequency dependent selection” in the context of macrotermitine symbiosis.

L. 107-108: Maybe these experiments on mutation-rate in other fungi should be slightly detailed to make clear why one should expect the same to happen in Termitomyces.

L. 113-143: Please, review this part to consider using the words “nest” or “mounds” when referring to the physical structure. For example, in line 114, perhaps use “nest” instead of “colony”. It would also be interesting to emphasize what provides the physical resistance to these structures, by shortly explaining how they are built.

L. 182: Stability of the fungal symbiont or stability of the mutualism? I believe this whole section refers to the latter.

L. 191-192: These questions really broke out the text. Consider removing them and leave only in table 2.

L. 299: typo? Perhaps, it should be “…it is the host’s…”

L. 346-347: replace “fungal mounds” to “fungus combs”.

L. 447: instead of “box 1” replace to “table 2”.

Figure 1: Nice figure! It summarizes very didactically the biology of the termite-fungus association and the main conflicts involved. Please, add italics in “Termitomyces” in the figure legend.

Author Response

Referee #1

Summary and overview

This manuscript seeks to explore how the mutualistic association between fungus-growing termites and fungal crops persists over the evolutionary time and short-term ecological time. Overall, the review systematizes current knowledge on conflicts between symbiotic partners that underlines the evolutionary stability of this mutualism and also brings pieces of information regarding fungus-growing termite colony longevity. It is a relevant contribution to the field of fungus-growing termite symbiosis, because it teases apart two aspects of the mutualism that ecologists and biologists wonder about. When dissecting these two aspects, the manuscript brings up open questions scattered in the literature and summarizes them in one specific manuscript. The references presented in this study are comprehensive and grasp the major aspects of the theme. The text needs some structural changes to improve its readability (see points #3 and #4). The comments below are mostly centered in the presentation of the text but some are also related to specific scientific issues that are not clear. Hope the suggestions help to improve the manuscript.

We are pleased to read the positive feedback of the first reviewer and thank the reviewer for his comments on the structural changes and scientific issues to improve the quality of the manuscript.

Major comments

  1. Evolutionary stability of mutualisms is a comprehensive area of research. While is understandable that this manuscript tries to limit the discussion on the target symbiosis (i.e. fungus-growing termites), however, when exploring how mutualistic associations can be stable, there are far more examples (i.e., other fungus-growing insects) available that could be explored in L. 43-56.

We agree that far more examples of stable symbioses exist than fungus growing termites. In addition to our example about the symbiosis between corals and their bacterial symbionts, we have now added a another example of a stable symbiosis to show the reader of the diversity of stable symbioses (lines 45-46). In addition to that we have moved the termite-fungus symbiosis to the examples (line 49-50), after which we continue with why we focus on this particular symbiosis.

  1. Although the manuscript brings relevant contribution to the field (see above), this is not clearly stated in the introductory paragraphs. Questions like (i) why this review is timely? and (ii) why is it so important to systematize the literature? are not obvious when reading the introduction section of the manuscript. This review would benefit if the rationale is clearly stated.

We have addressed our rationale more clearly in the last paragraph of the introduction by stating the fact that the symbiosis is widely studied, but that many gaps in our knowledge remain. We hope that this is satisfactory to the reviewer (lines 112-121).

  1. While it is a nice idea organizing the open research questions in one paper, the manuscript would also benefit if it makes one step forward. Instead of only listing the open questions in Table 2, an extra column should be added in this table to include potential approaches on how to address these questions. There is no need to fill every gap with proposed approaches, but there are approaches that may be available to pursue some of the posed questions.

The suggestion from reviewer 1 to include potential approaches to table 2 is very valuable and we agree that this is a very good contribution to our review. We have added an extra column to our review to include the potential approaches (line 516: table 2).

  1. Consider to compile all questions in table 2 and remove them from the main text (kept only when essential). It would also be interesting to clarify which aspects of the termites’ symbiosis suggest these questions as relevant, i.e., considering the ecology of fungus-growing termites, why one should expect such phenomenon to happen? Avoiding the overlapping between the content in table 2 and the main text would render a more straightforward text.

We have removed the redundant questions from the text, and only refer to table 2 where we originally addressed these questions (lines 210-211, 276-277, 370-374, 461-463).

  1. By the examples given at the intro, it sounds like the manuscript would also address the resilience of the colonies as an ecological entity, i.e., how termites’ colonies defend themselves from pathogens and predators, survive to environmental variables (as variations in food availability, temperature, humidity) during a colony or population life-span? Thus, it would be interesting to better delimitate the concepts of “short-term” resilience and “long-term” stability, and also explain how these concepts are applied to termites’ symbiosis. Differentiating: i) short-term resilience as adaptations of the colony to recover from perturbations as antagonists, competitors, and environmental variations, and ii) long-term stability as the stability of the mutualism on an evolutionary scale, exemplifying these definitions properly, would probably improve the text clarity.

We thank the reviewer for his comments, and agree that we only discuss resilience and inertia in the beginning of the introduction while this is not the focus of the rest of the manuscript. To better align the content of the introduction with the content of the rest of the manuscript, we have edited the start of the introduction by removing those terms and focussing on the stability of symbiosis without the separation between inertia and resilience (lines 26-34 and lines 50-59).

  1. It would be interesting to link the resilience of the mound physical structure to defensive advantages to the colony life-span (short-term resilience), as well as which features of the mound could confer such advantages.

We thank the reviewer for this feedback and the suggestion to include the resilience of the termite mound to the defensive advantages benefitting the colony lifespan. We have now included how the termite mound can protect the termite colony against predators and hostile environments and how the termites reduce erosion of the mound during the lifespan of a termite colony (lines 135-141).

  1. The conclusion section should close all gaps highlighted throughout the text, instead of only summarize the information already provided. As it is, the conclusion has several overlapping information with previous sections, but does not seem to recapitulate important aspects (as contextualizing resilience and stability) and to close the thought built during the review.

We had intended this section as a small summary of the manuscript. We have now changed the section heading to “summary” (line 471) and included more important aspects to recapitulate the important findings of our review (lines 473-515).

  1. It would be insightful to speculate how Termitomyces performance on different substrates could be correlated with short term resilience, as well as to link the fungal metabolic capacity to the conidia selection by workers. As such, it would be interesting to discuss whether strains that show a metabolic flexibility (i.e., are able to degrade multiple substrates and to survive by metabolizing several carbon sources) would/would not render a more stable association in short and long term. Also, it should be considered to conceive the possible influence of the microbiota on colonies metabolism and nutrition, and how it could affect the stability of the symbiosis.

We thank the reviewer for the suggestion to include a section about the impact of metabolic flexibility on the performance of the fungus. We have included this in the manuscript now (lines 266-272).

  1. Also very interesting would be to discuss the importance of resilience and stability to fungus-growing termites’ ecology by comparing them to other termites (non-fungus growing) and/or other fungus-growing insects. This would allow the authors to theorize possible impacts of the symbiosis on macrotermitine stability and resilience, and to relate this to the termites life-style and environment.

We agree that a comparison with other fungus-growing insects or non-fungus growing termites would be interesting. It would allow to investigate specific aspects of the termite-fungus symbiosis which are or not present in other symbiosis or non-fungus growing termites and how these would affect the stability of the symbiosis. However, we also feel that we cannot fully cover these comparisons within the scope of this review. A well performed comparison should involve a meta-analysis of many different symbioses.

Specific comments

The title seems not to cover the two aspects discussed in depth in the manuscript. Perhaps, something like “Stability of the mutualism and longevity of colonies of fungus-growing termites”.

We agree that we did not highlight the symbiosis enough in the title and focussed on the colonies. To include the main aspect of the manuscript; the stability of the symbiosis, we have changed the title into: “The Longevity of Colonies of Fungus-Growing Termites and the Stability of the Symbiosis” (line 2-3).

Consider use the term "conidia" for asexual spores, throughout the text.

This is a great suggestion, however we have decided to keep using the term “asexual spores” throughout the text to avoid confusion. Asexual spores of Termitomyces have been called various names (conidia, synnemata, nodules and mycotetes) (Rouland-Lefèvre et al. 2005).

  1. 21-22: Consider removing “fungus-growing termites” and “stability” because these keywords also appear in the title.

We have removed the keywords “fungus-growing termites”, “stability” and “symbiosis” (line 21).

  1. 24: remove “Introduction” and keep only one section heading.

We have removed “Introduction” from the section heading (line 24).

  1. 30: replace “inertia and resilience” to “these definition” to avoid repetition.

To reduce the focus on the terms intertia and resilience (see earlier comments), we have removed these sentences from the introduction (lines 30-34).

  1. 35-38: The explanation on host-symbiont association is not very clear.

We have added the definition of symbiotic associations to the introduction (line 28-29), and rewritten parts of the explanation on host-symbiont associations to make it more clear (lines 37-38).

  1. 43-56: correct for “P. kuna”.

We have changed the capital letter for a small letter (line 47).

  1. 59: How horizontal transmission occurs in the macrotermitine symbiosis is not clear at this point. It is indeed detailed below, but could cause confusion whether is not clearly stated at the introductory section.

We agree that the readers, at this point, are not informed how horizontal transmission can affect the stability if the symbiosis. We have therefor removed “horizontal transmission” from this point in the introduction to avoid a lengthy explanation here and/or a double explanation of the effects in the manuscript (line 65).

  1. 75: It would be helpful to clarify what is “positive-frequency dependent selection” in the context of macrotermitine symbiosis.

We have now clarified the term “positive frequency dependent selection” (lines 82-83).

  1. 107-108: Maybe these experiments on mutation-rate in other fungi should be slightly detailed to make clear why one should expect the same to happen in Termitomyces.

We have now shorty elaborated on the findings of these experiments (lines 119-121).

  1. 113-143: Please, review this part to consider using the words “nest” or “mounds” when referring to the physical structure. For example, in line 114, perhaps use “nest” instead of “colony”. It would also be interesting to emphasize what provides the physical resistance to these structures, by shortly explaining how they are built.

We have also changed the title of the section to “The stability of the termite Mound” (line 126).

In the section 2.1, with “colony” we also refer to the colony of termites, not the physical structure. When we refer to the physical structure we already use “mound”. However, we agree that it might be vague to write that the colony consists of the mound and the below-ground nest, so we have replaced the word “consists” into the word “occupies” (line 128).

In the section “Erosion and Persistence”, we have included how the mound is build and what provides the physical protection of these mounds to the colony (lines 135-141).

  1. 182: Stability of the fungal symbiont or stability of the mutualism? I believe this whole section refers to the latter.

The section refers to both the short term stability of the fungal symbiont within the termite mound as the long term stability of the mutualism. To clarify this, we have changed the section heading to “How Short-Term Stability of the Symbiont May Affect Long-Term Stability of the Symbiosis” (lines 201-201 and table 2).

  1. 191-192: These questions really broke out the text. Consider removing them and leave only in table 2.

We removed the questions from the text and only kept them in table 2 (lines 210-211).

  1. 299: typo? Perhaps, it should be “…it is the host’s…”

We thank the reviewer for spotting this. We have added “is” to the sentence (line 324).

  1. 346-347: replace “fungal mounds” to “fungus combs”.

Comb usually refers to only one fungal comb in one fungal chamber, while we meant that we only find one symbiont in the whole physical structure. We agree that fungal comb is not correct, and therefore we have changed “fungal mounds” into “termite mound” (line 376).

  1. 447: instead of “box 1” replace to “table 2”.

We thank reviewer 1 for also spotting this typo, a left-over from when the questions where in a box and not a table. We have changed “box 1” into “table 2” (line 477).

Figure 1: Nice figure! It summarizes very didactically the biology of the termite-fungus association and the main conflicts involved. Please, add italics in “Termitomyces” in the figure legend.

We thank the reviewer for this positive feedback on the figure. We have replaced the figure to a new version, without a typo (possitive, in box G). We have also italiced all “Termitomyces” in the figure legend (lines 87-111).

Reviewer 2 Report

This is a thorough and comprehensive review. The info-graphic style figures are stunning. In years of reviewing, I don't recall ever recommending a paper for acceptance without some suggestions for revision, until now. This paper could perhaps compare the phenomenon of longevity with fungus farming ants, but even without that comparison this review is solid. 

Author Response

Comments and Suggestions for Authors

This is a thorough and comprehensive review. The info-graphic style figures are stunning. In years of reviewing, I don't recall ever recommending a paper for acceptance without some suggestions for revision, until now. This paper could perhaps compare the phenomenon of longevity with fungus farming ants, but even without that comparison this review is solid.

We thank the reviewer for his kind words and his compliments on the figure. We absolutely agree that a comparison with other fungus farming insects would be interesting, however, as we also pointed out to reviewer 1, we also feel that it would require a systematic metanalysis to be able to perform such a comparison well.

Reviewer 3 Report

This paper nicely reviews what is known about the longevity and stability of colonies of fungus-growing termites, discussing both short and long-term factors susceptible to explain the maintenance of macrotermitine termites-Termitomyces symbiosis.

I find the manuscript well-structured, interesting, very clear and well-written. To the best of my knowledge, adequate key references were cited. I also find the figure 1 very useful and well-done. The authors end their paper with a long list of open questions which will certainly help future scientists to focus their research.  

I have only two minor suggestions:

(1) Could you add a paragraph introducing what is known about the taxonomy of both partners (i.e. number of genera and species) and their biogeography?

(2) As the fungus garden is a monoculture with very low genetic diversity, it should be highly susceptible to diseases. The latter may affect the stability of the system. Could you discuss how the termites cope with this potential issue?  

Other details are: 

  • Page 2: Plexaura kuna: write kuna with lowercase k.
  • Figure 1 (G): Replace "possitive" by "positive".
  • Figure 1: From my point of view the caption is too long and too redundant with the text already available in the figure itself. Perhaps you could simplify.
  • Caption of figure 1: Termitomyces should be written in italics. 

Author Response

This paper nicely reviews what is known about the longevity and stability of colonies of fungus-growing termites, discussing both short and long-term factors susceptible to explain the maintenance of macrotermitine termites-Termitomyces symbiosis.

I find the manuscript well-structured, interesting, very clear and well-written. To the best of my knowledge, adequate key references were cited. I also find the figure 1 very useful and well-done. The authors end their paper with a long list of open questions which will certainly help future scientists to focus their research. 

We thank reviewer 3 for the positive reaction to our manuscript and are very pleased to read that the reviewer thinks this will be useful for future scientist. We are grateful for the minor suggestions and have added these suggestions to our revised manuscript.

I have only two minor suggestions:

(1) Could you add a paragraph introducing what is known about the taxonomy of both partners (i.e. number of genera and species) and their biogeography?

We have added the taxonomic details of both partners into the introduction, as well as a description of their biogeography (lines 61, 62 and 66-68).

(2) As the fungus garden is a monoculture with very low genetic diversity, it should be highly susceptible to diseases. The latter may affect the stability of the system. Could you discuss how the termites cope with this potential issue? 

We agree that farming a monoculture makes the colony susceptible to diseases. We have included how termites reduce the pathogen load to keep their fungal cultures disease free (lines 347-352).

Other details are:

Page 2: Plexaura kuna: write kuna with lowercase k.

We have changed the capital letter for a small letter (line 47).

Figure 1 (G): Replace "possitive" by "positive".

We have replaced the figure to a new version, changing “possitive” into positive, in box G of the figure.

Figure 1: From my point of view the caption is too long and too redundant with the text already available in the figure itself. Perhaps you could simplify.

Caption of figure 1: Termitomyces should be written in italics.

We have also italized all “Termitomyces” in the figure legend (lines 87-111).

Round 2

Reviewer 1 Report

The manuscript now reads better and much improved. The authors did a very good job when preparing the updated version. The majority of the comments made by the reviewers were amended in the new version. Questions were entirely moved to Table 2, which is thoroughly filled with several insightful ideas directing future research on fungus-growing termites. This considerably improved the overall text, as well as the readability. In particular, all the content added under the topic “How Short-Term Stability of the Symbiont May Affect Long-Term Stability of the Symbiosis” is very interesting, widely discussed, and easy to follow.

L16: It would be better to replace for “termite colony lifespan”

L18: Leave just “stability of the symbiosis”

L19: “question on how to…”

L27: two or more species, since we have many examples of multipartite symbiosis in nature

L28: It is nice to mention the human microbiome here, because indeed it may “buffer” environmental alterations in the gut. However, this example should be explored to better explain how it is connected with the main issue of the paper. For instance, it could be explained how the microbiome buffers disturbances, linking to how the microorganisms associated with fungus-growing termites may buffer disturbances.    

L29-31: This is not the case for many symbiosis; in bacteria-bacteria or fungal-fungal interactions, for instance, the size of the interacting organisms is not necessarily an issue. This sentence could be rephrased by something like: “Host-microbial associations in most of the cases involve different-sized organisms. The host species tend to be the largest, that contain groups of symbionts…”

L33: Put in this way sounds like the host have some control over the symbiosis. It could be rephrased by something like: “The host tend to be favored by an optimization of the collective performance…”

L50: delete “today”…it seems a pending word in this sentence.

L54-58: It would be nice to add some information on the termites’ age polyethism to this sentence, as it has been discussed that the older workers forage for plant based substrates, while younger workers mix this material with spores and deposit this mixture at the comb.

L94-107: This last part of the intro could explicit the relevance of the review to better explain why this information compiled together may help termites’ research. For instance, it could be stated about the aim to address and compile major questions on this symbiosis, discussing future research possibilities for the field.

L119: correct to “saliva”.

L126: delete “created by termite colonies” because it is already implied the mounds are from termites.

L152: indefinite lifetime.

L180: change “colony\ies” to “colony (or colonies)”

L192-195: In this paragraph perhaps mention that you’ll also discuss topics 3.3. and 3.4, so that readers can follow the text better and they will be expecting for these topics.

L246-253: Good point, well discussed.

L287-288: It’s not very clear why the fungus tends to not be interested in alates production by the colony.

L292: cite Table 2 at the end of the sentence.

L325-327: A reference is missing to support this sentence.

L350: “Never” is a too strong term, consider to replace by “has not been observed”

L360: Consider to replace by “would not switch to another fungal”

L386-387: Is this already suggested in the literature? If it is not, then to rephrase to “We hypothesize that termites might have evolved additional…” could be better here.  

L388: Within the fungus comb?

L403-411: That’s very interesting

L445: “Discuss” instead of “ask”

Author Response

Reviewer 1

Comments and Suggestions for Authors

The manuscript now reads better and much improved. The authors did a very good job when preparing the updated version. The majority of the comments made by the reviewers were amended in the new version. Questions were entirely moved to Table 2, which is thoroughly filled with several insightful ideas directing future research on fungus-growing termites. This considerably improved the overall text, as well as the readability. In particular, all the content added under the topic “How Short-Term Stability of the Symbiont May Affect Long-Term Stability of the Symbiosis” is very interesting, widely discussed, and easy to follow.

We are very pleased to hear that the reviewer finds the manuscript much approved, and that our revisions were sufficient.

L16: It would be better to replace for “termite colony lifespan”

We have added “lifespan” in the sentence: “Both on the time-scale of a termite colony lifespan and..” (line 16).

L18: Leave just “stability of the symbiosis”

We have removed “of the mutualistic aspect” (line 18).

L19: “question on how to…”

We have added “on”, so the sentence now reads” “raises the question on how the mutualistic...” (line 19).

L27: two or more species, since we have many examples of multipartite symbiosis in nature

We agree with the reviewer that indeed also symbioses exist between more than two partners, so we have added “or more” (line 27).

L28: It is nice to mention the human microbiome here, because indeed it may “buffer” environmental alterations in the gut. However, this example should be explored to better explain how it is connected with the main issue of the paper. For instance, it could be explained how the microbiome buffers disturbances, linking to how the microorganisms associated with fungus-growing termites may buffer disturbances.   

We have now explored this example more and added that this is a long-term and stable symbiosis between the human body and the microbes (lines 28-30).

L29-31: This is not the case for many symbiosis; in bacteria-bacteria or fungal-fungal interactions, for instance, the size of the interacting organisms is not necessarily an issue. This sentence could be rephrased by something like: “Host-microbial associations in most of the cases involve different-sized organisms. The host species tend to be the largest, that contain groups of symbionts…”

We agree that indeed not all symbiosis involve larger and smaller species, and have now replaced “many” with “Host-microbe” (line 31).

L33: Put in this way sounds like the host have some control over the symbiosis. It could be rephrased by something like: “The host tend to be favored by an optimization of the collective performance…”

We have changed “the host has an interest in the maximisation” to “the host tend to be favoured by an optimization” (line 35-36).

L50: delete “today”…it seems a pending word in this sentence.

We have removed the word “today” (line 52).

L54-58: It would be nice to add some information on the termites’ age polyethism to this sentence, as it has been discussed that the older workers forage for plant based substrates, while younger workers mix this material with spores and deposit this mixture at the comb.

We have added information about which tasks are performed by which age class of termite workers (line 55-60).

L94-107: This last part of the intro could explicit the relevance of the review to better explain why this information compiled together may help termites’ research. For instance, it could be stated about the aim to address and compile major questions on this symbiosis, discussing future research possibilities for the field.

We have how addressed these points specifically in the last part of the introduction (lines 101-103).

L119: correct to “saliva”.

We thank the reviewer for spotting this typo, we have corrected “salvia” to “saliva” (line 120).

L126: delete “created by termite colonies” because it is already implied the mounds are from termites.

We have deleted “created by the termites” (line 126).

L152: indefinite lifetime.

We have changed “life” to “lifetime” (line 152).

L180: change “colony\ies” to “colony (or colonies)”

We have changed “\ies” to “(or colonies)” (line 180).

L192-195: In this paragraph perhaps mention that you’ll also discuss topics 3.3. and 3.4, so that readers can follow the text better and they will be expecting for these topics.

We have added an introduction to these topics to guide the reader through the paper (lines 193-197).

L246-253: Good point, well discussed.

We thank the reviewer for this compliment!

L287-288: It’s not very clear why the fungus tends to not be interested in alates production by the colony.
To make this more clear, we have more elaborately explained why it is not in the direct interest of the fungus that alates are formed. We hope our explanation is sufficient (line 288-291).

L292: cite Table 2 at the end of the sentence.

We now refer to table 2 (line 296).

L325-327: A reference is missing to support this sentence.

We added some references of known parasites, in particular of the genus Pseudoxylaria. Otherwise, mostly general references exist to support the statement that monocultures are susceptible to pathogens and that the yield might decrease, we have added such a reference (line 328-331).

L350: “Never” is a too strong term, consider to replace by “has not been observed”

We have replaced “is found” by “has been observed” to tone down the language (line 354).

L360: Consider to replace by “would not switch to another fungal”

We have followed this suggestion and replaced “will never” by “would not” (line 365).

L386-387: Is this already suggested in the literature? If it is not, then to rephrase to “We hypothesize that termites might have evolved additional…” could be better here. 

To our knowledge, such statements have not been suggested before in the literature, therefore we have added “we hypothesise that termites...” (line 389).

L388: Within the fungus comb?

We refer to all the fungal material within the termite mound, but it would indeed be clearer to refer to fungal combs. We have replaced “fungus” with “fungal combs” (line 391).

L403-411: That’s very interesting

We thank the reviewer for this comment, and are glad to hear that our hypotheses are found to be interesting.

L445: “Discuss” instead of “ask”

We have replaced “asked” with “discussed” (line 448).